# Experimental Study on the Evaluation of Physical Performance and Durability of Cement Mortar Mixed with Water Repellent Impregnated Natural Zeolite

**DOI:** 10.3390/ma13153288

**Published:** 2020-07-23

**Authors:** Chang Bok Yoon, Han Seung Lee

**Affiliations:** 1Architectural Engineering, Hanyang University, Seoul 04763, Korea; 2School of Architecture and Architectural Engineering, Hanyang University, ERICA, Ansan 5588, Korea

**Keywords:** natural zeolite, water repellent, impregnation, concrete durability

## Abstract

To complement the shortcomings of concrete surface treatment technology and improve the durability of concrete structure, the purpose of this study was to impregnate water-repellent performance into natural zeolite, which has many pores inside, to achieve water-repellent performance inside concrete. The physical performance and durability of cement mortar mixed with water-repellent natural zeolite was evaluated. Cement mortars were prepared by mixing ZWR1%, 3%, and 5% (ZWR: Zeolite + Water Repellent impregnation) in cement powder, and compressive strength, contact angle, water penetration test, resistance chloride penetration test, chloride diffusion coefficient, and accelerated carbonation test were evaluated. When the mixing ratio of ZWR increased, the compressive strength of the test specimen was reduced compared to OPC. In contact angle measurement, water penetration test, chloride penetration resistance test, chloride diffusion coefficient, and accelerated carbonation test, the ZWR-containing samples showed superior properties compared to OPC. It was found that the durability test results improved as the amount of mixing was increased, and the durability of the test specimen containing 5% ZWR was found to be the best.

## 1. Introduction

A reinforced concrete structure may face various deterioration factors depending on exposure to the external environment, and hence, its durability may decrease with the passage of time. One of the reasons for the deterioration is the penetration of water due to the microscopic pores in concrete, which induces corrosion of the reinforcing bars and results in a decrease in durability. Maintenance to prevent deterioration of concrete by moisture is becoming increasingly important for maintaining durability. The durability of concrete has emerged as a social problem; thus, research on the development of concrete with high quality and high durability has been gradually increasing, and various technologies have been adopted. Various admixture materials are used during the cement production process for the purpose of improving the durability performance of reinforced concrete structures and to reduce CO_2_ generation [1,2,3]. Through the development of these mixed materials, we try to prevent deterioration due to moisture and improve the durability of concrete. Techniques have also been developed to improve the durability of the concrete surface layer without providing another protective layer [4,5,6,7].

Such surface treatment techniques can be classified as either waterproof construction methods for blocking moisture, or a water-repellent construction methods for appropriately blocking the penetration of moisture while permitting ventilation of air. Silane-siloxane can also be used to coat the exposed surface of concrete in order to protect it from attack by deterioration factors. In this case, a water-repellent surface treatment for concrete needs to be used [8].

However, in the case of the surface treatment technique, it is not permitted to penetrate even into the micro pores of the concrete surface, forming a surface film. Hydrophobic surfaces repel water and, to some extent, other liquids [9]. It has the disadvantage that deformation such as deterioration and peeling easily occurs due to external influences (ultraviolet rays, freezing, etc.). This method may not be appropriate, as the rebuild cycle becomes shortened. As a result, this method can be costly and time consuming, as well as reducing the durability of concrete [10].

Water-based water repellent, which is a material that imparts water repellency to the structure, is a solution in which siloxane is dissolved in water. Applying a hydrophobic agent makes the concrete surface water-repellent [11]. In existing papers on surface treatment technology, methods for impregnating concrete surfaces in order to reduce water permeation have mainly been studied [12]. The experimental method in this paper is an experiment that imparts water repellency inside concrete. Physical and durability performance experiments were conducted through experiments in which the material was impregnated with a water-repellent component and added when mixing the concrete. When it is mixed directly to provide water repellency inside concrete, material separation occurs due to the different specific gravity between the water repellent and the concrete, and heterogeneity can occur. As a result, the possibility of decreasing the water repellency cannot be excluded when material separation occurs. Furthermore, it is difficult to ensure a uniform dispersion of the water-repellent component in the concrete, and it is necessary to take measures against the problem that occurs when the silane-siloxane is mixed into the concrete.

In contrast, since the 1980s, much research has been conducted on the durability of concrete impregnated with silane-based water-repellent agents [13]. Impregnation is considered to have a high possibility of using natural zeolite as an operation for inserting a substance into a porous body in a gas or liquid state and improving the properties of the object according to the purpose of use. Natural zeolite has many pores inside the particles, as a pozzolan-based natural component [14], and it can be expected to play a role as a water-repellent carrier by using this. When using high-quality natural zeolite as an admixture for concrete, the pozzolan reaction of natural zeolite starts from early hydration, maintaining compressive strength and increasing the long-term strength, reducing speculation, and suppressing the alkali-aggregate reaction. It has been reported to play a role in improving physical properties [1,15]. Studies have been conducted using natural zeolite as an alternative material for cement admixtures [15].

Therefore, the purpose of this study was to introduce water repellency inside the concrete, improving the durability of the concrete by arresting the moisture penetration inside the micro pores [16]. A natural zeolite rich in porosity was impregnated with water repellency and mixed into the interior of cement mortar to evaluate physical properties and durability performance.

## 2. Materials and Specimens

### 2.1. Materials

Ordinary Portland cement (OPC) of ASTM C 150 [17,18] with a density of 3.15 g/cm^3^ (S Company, Seoul, South Korea) [17] and 3000 cm^2^/g (Blaine) was used in this experiment. Table 1 shows the chemical composition in percentage. The domestic fine aggregate specified in KS L ISO 679 was used. The natural zeolite used for the test was mined from Pohang Gyeongbuk-do, Korea and had a specific surface area of 15600 cm^2^/g (Blaine)and density of 2.3 g/cm^3^. Table 2 shows the chemical composition (in percentage) of natural zeolite analyzed by X-ray fluorescence analysis in accordance with the method specified in KS E 3076 [18]. Table 3 shows the percentage of harmful components present in natural zeolite. 

The water repellent used in the experiment was silane-siloxane-based with a solid content of 50%, and the pH was found to be 12. Table 4 shows the proportion of the mixture of diluent and water-based repellent.

### 2.2. Mortar Specimens Mix Proportion

For this experiment, four types of mortar specimens were prepared according to the mixing ratio by adding water-repellent impregnated natural zeolite to the mortar [17]. OPC 100%, water-repellent impregnated natural zeolite replacement rate 1%, 3% and 5%; W/B was fixed at 40%. The cement paste and mortar were mixed by the method of KS L 5109 “practice for mechanical mixing of hydraulic cement pasted and mortar of plastic consistency”.

Natural zeolite has many micro pores inside, so it can contain moisture between the pores. The presence of zeolite can retard the hydration process, thereby reducing the permeability, sorptivity and diffusivity of concrete, because it reduces porosity and improves the transition zone structure between the blended cement paste and the aggregate [19,20,21,22,23]. By drying the natural zeolite at 100 °C for 24 hours and soaking it in the water repellent for 48 hours, the active ingredient of the water repellent sufficiently penetrates into the pores of the natural zeolite. Then, after being sufficiently dried at 100 °C for 24 hours, it was pulverized to prepare water-repellent natural zeolite powder. The dried water-repellent zeolite was pulverized and then passed through a 300 μm sieve for use. Table 5 shows the mix proportion. Figure 1, below, shows photographs of natural zeolites that are water repellent.

### 2.3. Specimens

#### 2.3.1. Specimen for Compressive Strength, measurement of Contact angle and Mercury Intrusion Porosimetry (MIP) 

The specimens for measurement of compressive strength, contact angle and mercury intrusion porosimetry (MIP) were produced in a square mold of 50 mm × 50 mm × 50 mm [24]. Figure 2a,b shows the appearance of the specimens.

#### 2.3.2. Specimen for Chloride Ion Migration Coefficient and Evaluation

Mortar was compounded, and this was put in a Ø100 × 200 mm cylinder and sealed to prepare a test specimen. After 24 hours, the test specimen was released from the mold and subjected to underwater curing. After curing, it was cut into Ø100 × 50 mm depending on the test day, and the two central samples in the middle were used. Figure 3 shows the appearance of the Chloride Ion Migration Coefficient and Evaluation specimen.

#### 2.3.3. Accelerated Carbonation Test

The test for evaluating the carbonation depth was carried out based on KS F 2584 [25], the standard test method for accelerated carbonation of concrete. A 100 mm × 100 mm × 400 mm specimen was prepared and cured in water for 28 days, and then the remaining surface excluding the CO_2_ permeation surface was coated with epoxy and sealed. Figure 4 shows the appearance of the specimen [25].

#### 2.3.4. Curing Method

To compare the chloride penetration resistance and the compressive strength of the concrete test pieces according to the curing conditions, concrete test specimen were prepared and demolded after 24 hours. The mortar specimens were subsequently cured in water for 7, 28, 56, and 91 days. Subsequent samples were completely immersed in water at a temperature of 20 ± 2 °C for underwater curing.

## 3. Experimental Method

### 3.1. Mercury Intrusion Porosimetry (MIP)

Mercury Intrusion Porosimetry (MIP) is one of the widely used methods for studying the pore structure characteristics of cementitious materials, in which mercury is injected into a sample and the volume is measured by the amount of mercury infused. The test specimens for microanalysis were immersed in acetone in order to uniformize the analysis of the experiment, and the hydration reaction was stopped by curing days. After that, it was completely dried at a temperature of 50 °C, pulverized and analyzed. Figure 5 shows the appearance of the state of the experiment of MIP [26]. 

### 3.2. Evaluation of Specimen Compressive Strength

The measurement of the compressive strength of cement mortar is evaluated based on KS L 5105 [27] and ASTM C39 [24] after 7, 28, 56, and 91 days of hardening. 50 mm × 50 mm × 50 mm square mortar specimens at each experimental level were made. Three test specimens were measured simultaneously for the compressive strengths of mortar, and the average values thereof were used [25].

### 3.3. Evaluation of Contact Angle

Generally, solid surfaces having a contact angle less than 90° are considered to be hydrophilic and solid surfaces having a contact angle of 90° or more are considered to be hydrophobic [28,29]. The contact angle was measured by the sessile drop method. The contact angle by sessile drop method is mainly measured using large optical instruments such as microscopes. It can be measured with a minimal amount of liquid, and very small contact angles can also be measured [28,29]. Figure 6 shows the specimen and the test method and the contact angle data obtained are presented in Table 6.

### 3.4. Evaluation of Chloride Ion Migration Coefficient

Exposure tests are the most accurate way to evaluate the resistance of concrete to chloride ingress. However, the exposure test takes a very long time. Therefore, it is common to use electrochemical acceleration methods to evaluate the resistance and diffusion coefficient of harmful ions in concrete. Generally, the ASTM C 1202 [30] and NT-BUILD 492 [31] methods are used to assess chloride penetration of concrete. In this study, among the various methods for assessing chloride ion permeation resistance, the chloride transfer coefficient from anomalous transfer experiments is a quantitative assessment method and is often used in the Nordic method. It was carried out by the method of NT BUILD 492, which is the regulation [31]. Figure 7 shows a schematic of the cell configured for this experiment and a photo of the experimental setup installed. The measurement parameters are shown in Table 7.

This experiment is an unsteady electrophoresis experiment to find the chloride transfer coefficient of a repair material composed of concrete, mortar, and cement. A mortar test specimen with a size of Ø100 mm × 50 mm was prepared, and a desiccator was filled with a saturated Ca(OH)_2_ solution, and the inside was saturated in preparation. Epoxy coating was performed on the rest of the parts except the penetration surface, so that chloride ions could penetrate only in one direction. The positive electrode was fully filled with 0.3 M NaOH and the negative electrode was filled with 10% NaOH solution. Then, the initial current value (I_30_V) was measured, and the actual voltage according to the measurement was adjusted according to the initial current value in Table 7. The chloride ion permeation resistance was tested by using a potential difference by selecting a predetermined time according to the electric current. After the test, the test specimen was divided into two in the axial direction, and a 0.1N AgNO3 solution was sprayed onto the divided portion.

Thereafter, the discolored silver portion of the test specimen was determined as the penetration depth of chloride. The diffusion coefficient was calculated by determining the average of seven measurements at 10 mm intervals of the chloride penetration depth [17,30,31,32,33]. Equation (1), below, is used to estimate the diffusion coefficient [17,30,31,32,33].
(1)$$D_{nssm}=\frac{RT}{zFE}\frac{x_{d}-\alpha \surd x_{d}}{t}$$
(2)$$With\;:\;E=\frac{U-2}{L},\;\;\alpha =2\sqrt{\frac{RT}{zFE}}erf⎺1\left(1-\frac{2Cd}{Co}\right)$$
where
*D_nssm_*—is a non-steady-state migration coefficient (m^2^/s)*R—*gas constant, R = 8.314 J(K·mol)*F—*faraday constant = 9.648 × 10^4^ J(V·mol)*U*—absolute value of the applied voltage (V)*T*—average of the initial and final temperatures in the anolyte solution (°C)*L*—thickness of the specimen (mm)$x_{d}$—average value of the penetration depths (mm)*t*—test duration (hour)*C_d_*—chloride concentration at which the color changes*C_o_*—chloride concentration in the catholyte solution [17,32,33,34]

### 3.5. Accelerated Carbonation Test

The penetrated test specimens were placed in a chamber under the conditions of relative humidity of 60 ± 5%, temperature of 20 ± 2 °C and CO_2_ 5 ± 0.1% to allow CO_2_ to penetrate. The depth of oxidation was measured by decomposing a position 60 mm away from the end of the test specimen, and spraying 1% concentration phenolphthalein solution on the cut surface to measure a section where the color changed to red purple, using calipers from the surface. After that, each of the average values were calculated. The carbonation penetrate depth was measured at 1, 4, 8 and 13 weeks [25].

### 3.6. Water Penetration Test

To confirm the water resistance of each test specimen, the water permeability test method specified in KS F 4919 was followed, and cement mixed polymer waterproof material [34] was used. This standard is specified for waterproofing materials based on polymer admixtures and cement-based hydraulic inorganic powders, which is not compatible with this study, but was used to incorporate water repellents into cement mortar. 

The test was conducted according to the criteria for mixing the polymer repellent in a situation where the criteria for mixing and using the water repellent in the concrete are somewhat ambiguous. After completely drying a Ø100 mm × 50 mm circular test specimen, the side surface was coated with an epoxy resin and sealed in order to allow water to permeate only in both directions. After that, the weight was measured before the water penetration test, and after applying a water pressure of 0.3 N/mm^2^ to the water penetration test device connected to the air compressor for 3 hours, the test specimen was separated, lightly wiped, and the weight after the water penetration test was measured. The absorption rate was measured, and then the wetted surface was checked with the eye when infiltrating the test specimen with the water in the center. Figure 8 shows the water penetration test set-up.

## 4. Results and Discussion

### 4.1. Mercury Intrusion Porosimetry (MIP)

In this study, mercury was injected under the condition of contact angle of 130° up to 430 MPa using the mercury injection method. The experimental results of MIP were measured after stopping the hydration reaction according to the curing days of the material, as follows. It was confirmed that the size of pores decreased with curing days of all specimens. Table 8 shows porosity measured by MIP(28, 91) data. As a result of the measurement after 91 days, it was confirmed that the porosity decrease of ZWR1% was 88.8%, ZWR3% was 80.7%, and ZWR5% was 64.4% based on OPC. The pore distributions at 28 and 91 curing days are presented in the first graph of Figure 9. It was observed that the test specimens using ZWR were distributed mainly within 50 nm. Cumulative pores at 50 nm and 10 nm were confirmed at 0.05–0.06, and even large pores up to 1,000,000 nm and 100,000 nm could also be confirmed in the OPC with the most pores. The results of pore distribution at 91 curing days are discussed in the following paragraph. In the test specimen using ZWR, it was confirmed that the distribution was mainly between 30 nm after 91 days, and in the case of OPC, it was distributed between 50 nm and 100 nm. Cumulative pores from 10 nm to 50 nm were confirmed at 0.04–0.05, and even large pores up to 1,000,000 nm and 100,000 nm were able to be seen, with the most cumulative pores being in OPC. In the ZWR5% test specimen, pores having a size of 100,000 nm to 10,000 nm could not be found.

### 4.2. Result of Concrete Compressive Strength

Table 9 shows the compressive strength results of cement mortar incorporating 1%, 3%, and 5% of ZWR, a natural zeolite impregnated with water repellent, into the amount of cement, respectively. For convenience, the same compression data are presented in graphical form in Figure 10. It was confirmed that the results of compressive strength measured at 7, 28, 56, and 91 days were in the order of OPC > ZWR1% > ZWR3% > ZWR5%. Compared to OPC, ZWR1% showed 90% strength, ZWR3% showed 86% and ZWR5% showed 82% compressive strength. 

Therefore, it was confirmed that the compressive strength increased as the curing time increased, but the rate of increment in the compressive strength decreased as the ZWR content increased. ZWR acts as a cushion of the silane-siloxane water-repellent component inside the test specimen, and it is considered that the compressive strength slightly decreased depending on the amount of the mixture.

### 4.3. Result of Evaluation of Contact Angle

Figure 11 and Figure 12 show the measurement results of the water-repellent contact angle after 7, 28, 56 and 91 days. The measured contact angle values are tabulated and presented in Table 10. The measured results were in the order of ZWR5% > ZWR3% > ZWR1% > OPC.

The measurement of the contact angle for OPC and ZWR1% samples was not possible. However, for ZWR3% and ZWR5% samples the obtained values were 74° and 93°, respectively. In the case of ZWR5%, the sample hydrophobicity value exceeded 90°. The OPC and ZWR1% specimens showed almost no contact angle expression, and the ZWR3% and ZWR5% specimens developed surface tension due to the incorporation of the water-repellent component, and the contact angles increased. This observation may be due to hardening of the mortar voids and voids had been filled with hydration products and ZWR. The variation of contact angles with the curing time have been presented graphical form in Figure 12. The results indicate that, as the mixed amount of ZWR increases, the contact angle tends to increase and the compressive strength tends to decrease. Therefore, it is necessary to examine the correlation of the mixed amount of ZWR to achieve the optimum properties.

### 4.4. Evaluation of Chloride Ion Migration Coefficient

The chloride ion migration coefficient data obtained from this test are tabulated in Table 11. The plot from the obtained data is presented in Figure 13a. After the test was completed, the test specimens were split in the axial direction, sprayed with a 0.1 N silver nitrate (AgNO_3_) solution [35,36,37] on the cross-section, and when dried, the discolored part of the test specimen was displayed according to the penetration depth of chloride ions [35,36,37]. 

The appearance of the surface of the broken samples after chloride ion migration coefficient tests is presented at the right side of Figure 13b. The diffusion coefficient values were obtained by measuring seven points at intervals of 10 mm and setting the average value of the depths as the chloride penetration depth [32]. In the test specimen using ZWR, it was confirmed that the permeation of chloride was surely reduced [12]. OPC showed a 13.3 × 10^−12^ m^2^/s diffusion coefficient as a result of the chloride diffusion test on the 28th day of age, whereas for ZWR1%, ZWR3%, and ZWR 5%, the obtained values for the same age were 13.1 × 10^−12^ m^2^/s, 8.8 × 10^−12^ m^2^/s, and 7.39 × 10^−12^ m^2^/s, respectively. The 91 day diffusion coefficient showed the same tendency as on the 28th, with OPC appearing at 13.1 × 10^−12^ m^2^/s, followed by ZWR1% 9.51 × 10^−12^ m^2^/s, ZWR3% 6.36 × 10^−12^ m^2^/s, ZWR5% 4.77 × 10^−12^ m^2^/s. 

Moisture penetrates into the voids of the test specimen and moves to the inside, and the water penetrates through the mechanism by which chloride also moves. From the experimental results, it can be confirmed that the diffusion coefficient tended to decrease more and more depending on the mixing amount. It is considered that the distribution and size of the voids were reduced, and that is why the ZWR powder pushed out the penetration of water into the pores, improving the resistance performance against chloride.

### 4.5. Accelerated Carbonation Test

All test specimens were easily subjected to a carbonation test 7 days, 28 days, 56 days, and 91 days after being cured in water for 28 days. The concentration of CO_2_ was fixed at 5%. Carbonation is a phenomenon in which a trace amount of CO_2_ gas in the atmosphere reacts with a hydration product in concrete to generate calcium carbonate, which causes the destruction and corrosion of the passive film of the reinforcing steel due to the decrease in alkalinity inside the concrete [37]. 

The carbonation process of concrete is greatly affected depending on the relative humidity inside the pores [37]. The experimental results of the accelerated carbonation test are presented in Table 12. It was found that the carbonation depth increased in all specimens with increasing curing days. There were differences in the penetration depths depending on the mixing ratios of ZWR. The depths of carbonations were in the order of OPC > ZWR1% > ZWR3% > ZWR5%. 91 days penetration depth was OPC: 5.8 mm, ZWR1%: 5 mm, ZWR3%: 4.6 mm, ZWR5%: 3 mm. The ZWR in between the pores also showed resistance to CO_2_ penetration. The ZWR mixed test specimen had a lower porosity and higher water tightness than OPC, so it was judged that ZWR retained water resistance and speculative resistance inside the pore. Figure 14 shows a graphical representation of the CO_2_ penetration depth of the samples with different curation ages.

### 4.6. Water Penetration Test

To confirm the water resistance of each test specimen, an experiment was conducted on the 28th day and the 91st day according to the water penetration test method specified in KS F 4919 “Cement-containing polymer waterproof material”. After measuring the weights before the water penetration tests and applying a water pressure of 0.3 N/mm^2^ to the water penetration test device connected to the air compressor for 3 h, the test specimens were separated, lightly wiped, and the weights after the water permeation tests were measured to check the water absorption. Table 13 and Table 14 and Figure 15 show the results of the following water penetration tests. 

After the water pressure of 0.3 N/mm^2^ was applied for 3 h, the test specimens were separated, and the weights after the water penetration test were measured to confirm the absorption rate. The penetration depths were found to be in the order of the OPC > ZWR1% > ZWR3% > ZWR5%. This result confirms the increase in the permeation resistance of water in the ZWR-added samples. It seems that the hydration product and ZWR filled the micro pores according to the progress of the treatment, and the water tightness was promoted in the test specimen. As the amount of ZWR mixed in increased, the penetration resistance performance for absorbing water improved.

## 5. Conclusions

To improve the durability of concrete, this study focused on the physical performance and durability of cement mortar [38] mixed with water-repellent natural zeolite. ZWR, a natural zeolite with water-repellent properties, was mixed in the cement powder with different proportions of 1%, 3% and 5%. Compressive strength, contact angle, water penetration test, chloride penetration resistance, chloride diffusion coefficient, and accelerated carbonation tests were conducted. The key observations of this study are as follows.

The evaluation results of contact angle and compressive strength for the ZWR 5% specimen showed a compressive strength of 82% of a standard mortar and a contact angle more than 5 times. The ZWR-incorporating specimens appeared to have increased air content due to the effect of some silane-siloxane particles not bound to the cement particles. It is considered that the cement and mortar sand and paste were delayed in bonding, the adhesive strength was reduced, and the compressive strength was also reduced. In addition, it is considered that the ZWR-filled sample has excellent moisture permeation resistance, but its compressive strength decreases due to the cushioning effect.It was confirmed that the penetration resistance performance of water for the ZWR 5% test specimen was the best among the test specimens. As the curing days of the material passed, the hydration product and ZWR were filled into the pores to increase the water tightness, and the greater the amount of ZWR mixed in, the more the resistance to water absorption increased.In the chloride penetration resistance test and chloride diffusion coefficient tests, it was observed that the penetration and migration of chloride-containing water was higher in the test specimen containing ZWR than in OPC. In comparison to OPC, it was found that generally, the amount of charge and the diffusion coefficient due to the penetration of water from the specimen mixed with ZWR were inversely proportional.The MIP test results showed that the pore size and cumulative pore tended to decrease with curing days. In the case of the test specimen containing ZWR, it is considered that the hydration product of cement has the same effect as that of filling the pores in the interior and reducing the pores size and the pore ratio. In addition, compared with OPC, the size of the pores and the cumulative pores of the specimen with ZWR 5% were about 64%. Based on the test results, the increase of contact angle could be evaluated due to the decrease of chloride diffusion coefficient [39], the decrease of CO_2_ penetration depth in the carbonation accelerated test, and the decrease of water absorption.It was confirmed that the greater the amount of ZWR mixed, the more the compression strength tended to decrease, but the durability performance improved. It can be judged from this experiment that the optimum ZWR mixing ratio for improving the durability of concrete due to the penetration of water and imparting water repellency to the inside of concrete is 5%.It is considered that additional physical and durability experiments are needed to evaluate the resistance to moisture penetration inside the concrete.

## Figures and Tables

**Figure 1 materials-13-03288-f001:**
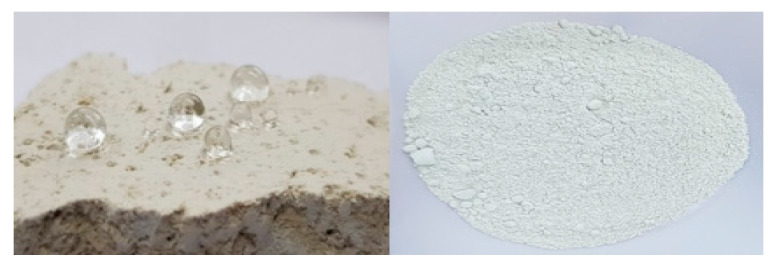
Water-repellent impregnated natural zeolite.

**Figure 2 materials-13-03288-f002:**
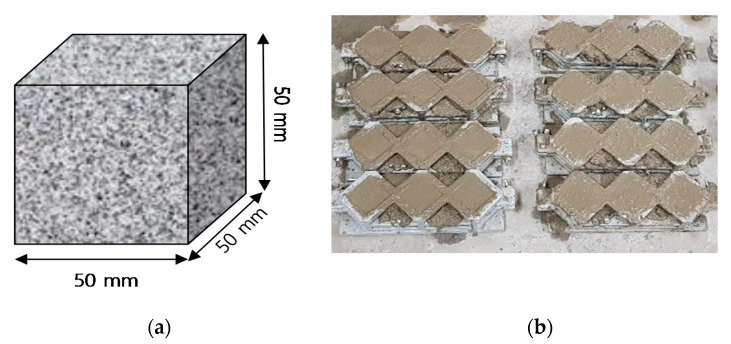
(**a**) Schematic diagram of Specimen; (**b**) 50 mm × 50 mm × 50 mm square cement mortar production.

**Figure 3 materials-13-03288-f003:**
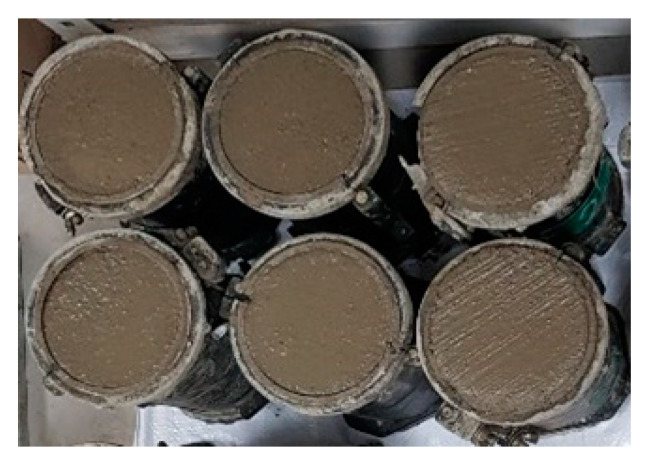
Specimen for chloride ion migration coefficient.

**Figure 4 materials-13-03288-f004:**
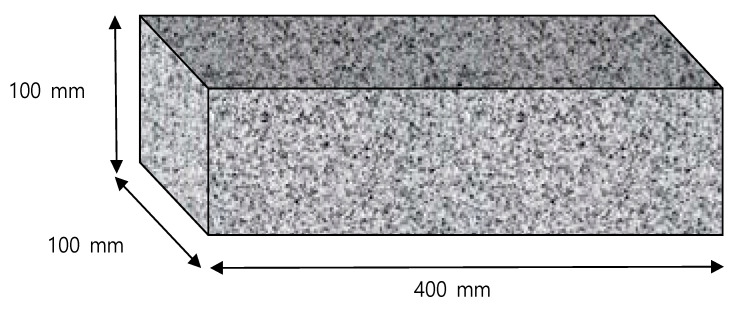
Specimen for accelerated carbonation test.

**Figure 5 materials-13-03288-f005:**
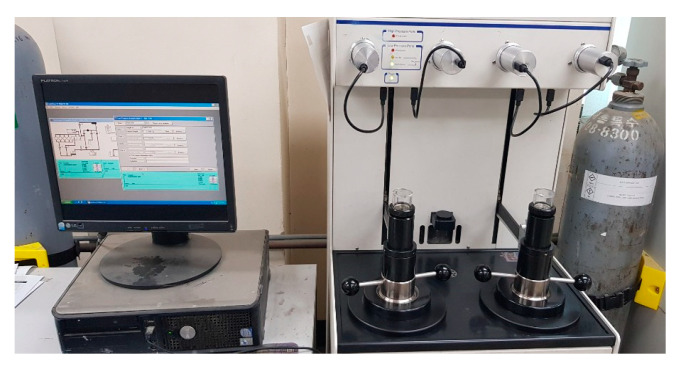
Mercury intrusion porosimetry test set-up.

**Figure 6 materials-13-03288-f006:**
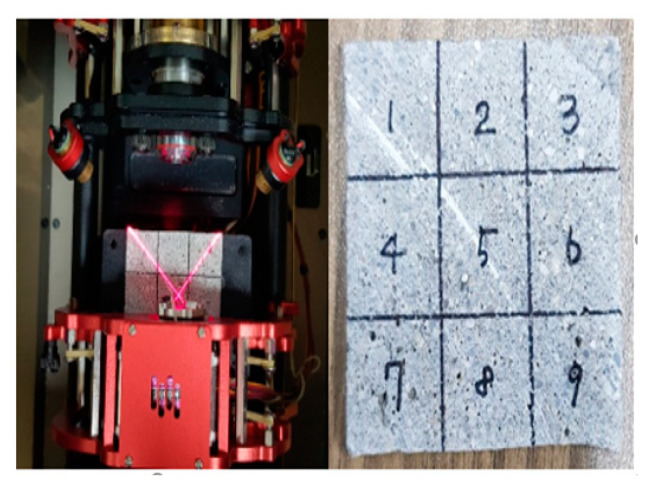
Specimen for contact angle measurement test.

**Figure 7 materials-13-03288-f007:**
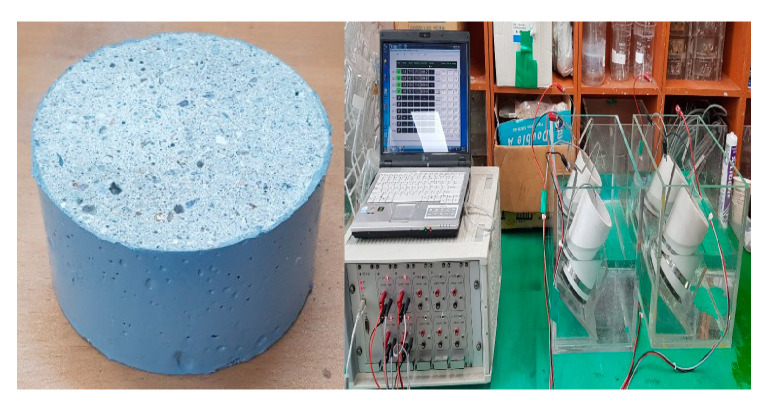
NT BUILD 492 test set-up.

**Figure 8 materials-13-03288-f008:**
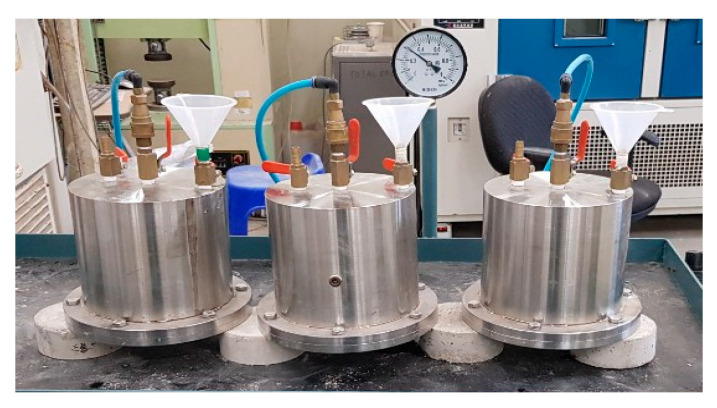
Water penetration test set-up.

**Figure 9 materials-13-03288-f009:**
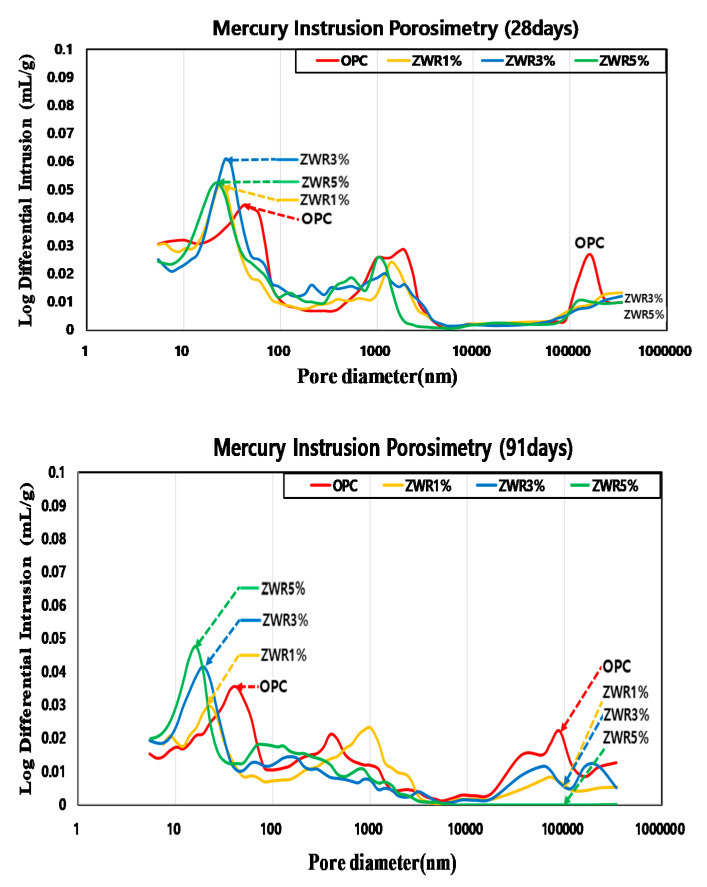
Results of MIP graph (28, 91 days).

**Figure 10 materials-13-03288-f010:**
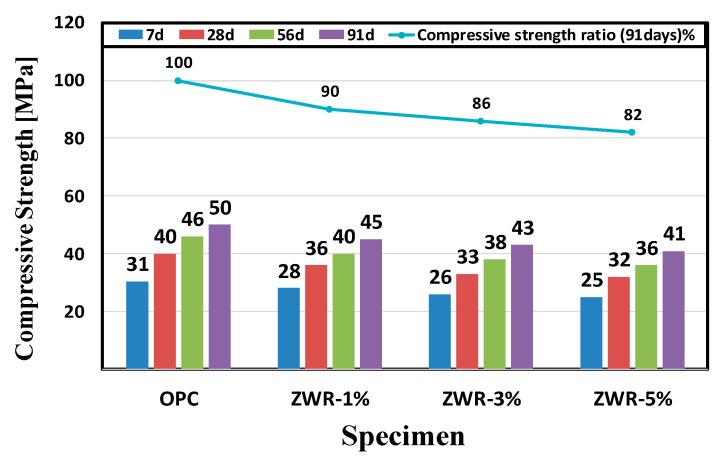
Graph of compressive strength test.

**Figure 11 materials-13-03288-f011:**
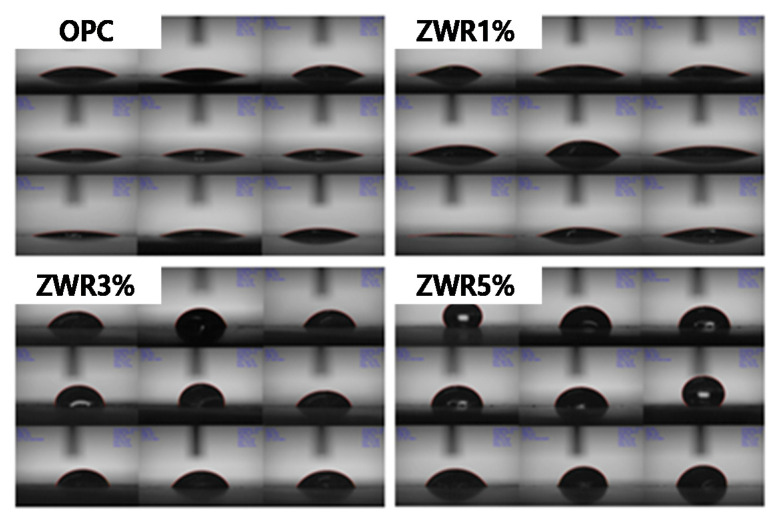
Result of contact angle test 91 days.

**Figure 12 materials-13-03288-f012:**
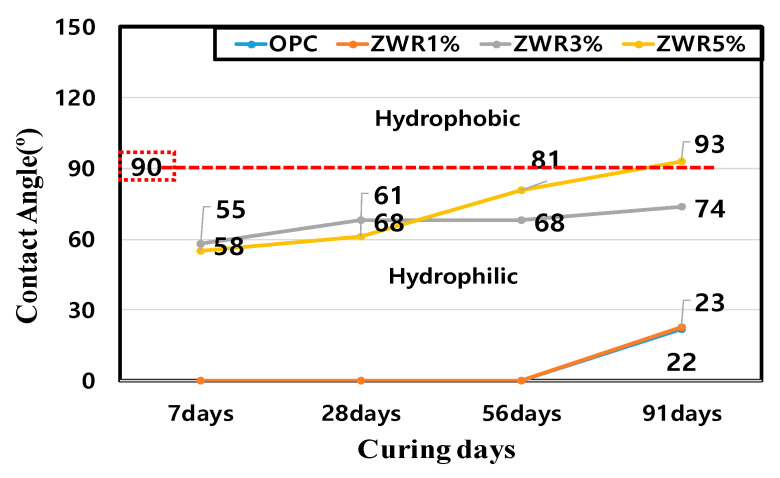
Graph of contact angle measurement with curing time.

**Figure 13 materials-13-03288-f013:**
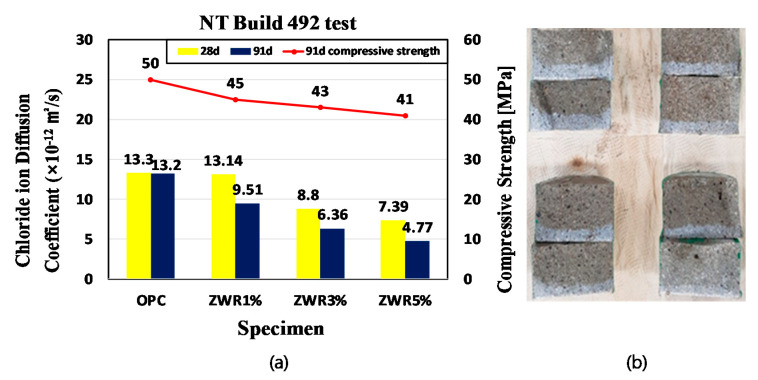
(**a**) Graph of evaluation of resistance to chloride ion penetration test; (**b**) Cross-section of the test specimen after chloride penetration.

**Figure 14 materials-13-03288-f014:**
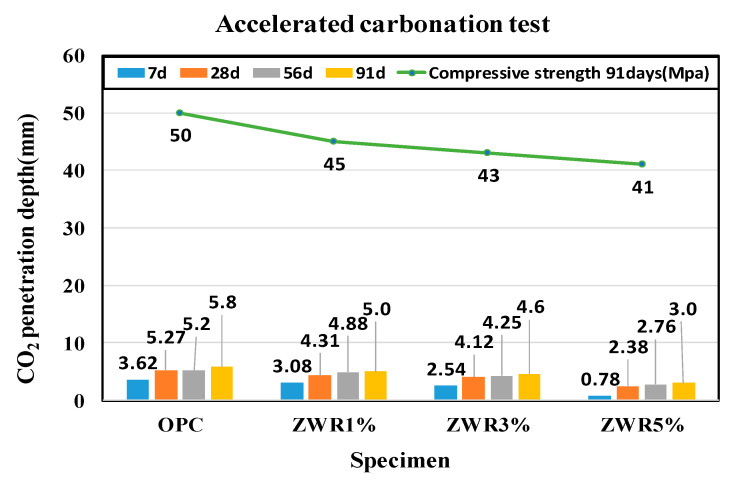
Graph of accelerated carbonation test.

**Figure 15 materials-13-03288-f015:**
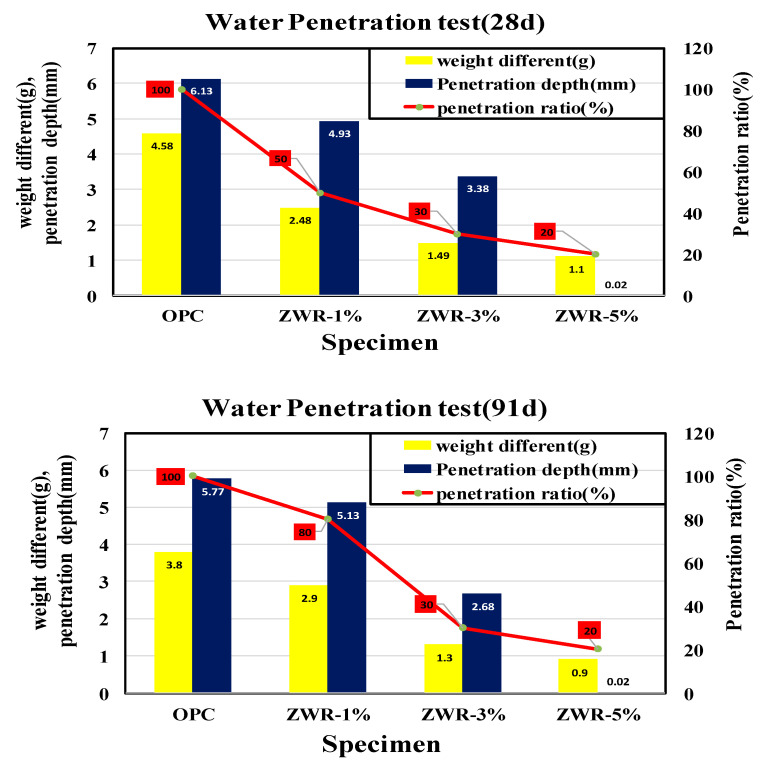
Graph of water penetration test.

**Table 1 materials-13-03288-t001:** Chemical compositions of cement.

Name	Chemical Compositions (%)									
	SiO_2_	Al_2_O_3_	TiO_2_	Fe_2_O_3_	CaO	MgO	SO_3_	K_2_O	Etc. /Lg. Loss	L.O.I
OPC	19.74	5.33	0.30	2.93	61.74	3.78	2.47	0.89	2.82	2.3

**Table 2 materials-13-03288-t002:** Chemical compositions of natural zeolite.

Name	Chemical Compositions (%)						
	SiO_2_	Al_2_O_3_	Fe_2_O_3_	K_2_O	Na_2_O	CEC	Etc./Lg. Loss
Zeolite	66.8	13.2	1.68	3.02	1.16	106	14.14

**Table 3 materials-13-03288-t003:** Hazardous component test of natural zeolite.

Name	Chemical Compositions (mg/kg)							
	As	Cd	Hg	Pb	Cr	Cu	Ni	Zn
Limit	20	2	1	50	90	120	20	400
Result		0.64			7.87	4.37	3.75	23.81

**Table 4 materials-13-03288-t004:** Chemical compositions of water repellent.

Name	Chemical Compositions (%)					
	Color	Kind	Effective Ratio (%)	Diluent	Freeze Stability	pH
Water Repellent	White	Silane - Siloxane	50	Water		12

**Table 5 materials-13-03288-t005:** Cement mortar mix proportion.

Name	W/B (%)	Unit Weight (kg/m^3^)					
		C	W	Sand	Impregnation Admixture = ZWR		Addition Ratio
					Zeolite	WR^1^	
OPC	40%	510	204	1530			
ZWR1%	40%	510	208	1530	5.1	5.1	1%
ZWR3%	40%	510	216.2	1530	15.3	15.3	3%
ZWR5%	40%	510	224.4	1530	25.5	25.5	5%

**Table 6 materials-13-03288-t006:** Water penetration according to contact angle.

Surface Contact Angle (θ)	Permeability
>130°	Very good repellency
110–130°	Good repellency
90–110°	Slight wetting
30–90°	Pronounced wetting
<30°	Surface completely wet

**Table 7 materials-13-03288-t007:** Test voltage and duration for concrete specimen with normal binder content [17,30,31,32,33].

Initial Current I_30V_ (with 30V) (mA)	Applied Voltage U (after Adjustment) (V)	Possible New Initial Current I_0_ (mA)	Test Duration (h)
I_0_ < 5	60	I_0_ < 10	96
5 ≤ I_0_ < 10	60	10 ≤ I_0_ < 20	48
10 ≤ I_0_ < 15	60	20 ≤ I_0_ < 30	24
15 ≤ I_0_ < 20	50	25 ≤ I_0_ < 35	24
20 ≤ I_0_ < 30	40	25 ≤ I_0_ < 40	24
30 ≤ I_0_ < 40	35	35 ≤ I_0_ < 50	24
40 ≤ I_0_ < 60	30	40 ≤ I_0_ < 60	24
60 ≤ I_0_ < 90	25	50 ≤ I_0_ < 75	24
90 ≤ I_0_ < 120	20	60 ≤ I_0_ < 80	24
120 ≤ I_0_ < 180	15	60 ≤ I_0_ < 90	24
180 ≤ I_0_ < 360	10	60 ≤ I_0_ < 120	24
I_0_ ≥ 360	10	I_0_ ≥ 120	6

**Table 8 materials-13-03288-t008:** Porosity measured by MIP (28, 91days).

Name	Porosity (%)		OPC 91D Porosity Ratio (%)
	28 d	91 d	
**OPC**	16.2	13.5	100
ZWR1%	15.8	12.0	88.8
ZWR3%	12.8	10.9	80.7
ZWR5%	11.4	8.7	64.4

**Table 9 materials-13-03288-t009:** Result of compressive strength test.

Compressive Strength (MPa)					
Specimen	7 Days	28 Days	56 Days	91 Days	91 Days Ratio (%)
OPC	31	40	46	50	100
ZWR1%	28	36	40	45	90
ZWR3%	26	33	38	43	86
ZWR5%	25	32	36	41	82

**Table 10 materials-13-03288-t010:** Result of contact angle test.

Name	Contact Angle (°)			
	7d	28d	56d	91d
OPC				22
ZWR1%				23
ZWR3%	58	68	68	74
ZWR5%	55	61	81	93

**Table 11 materials-13-03288-t011:** Result of evaluation of resistance to chloride ion penetration.

Name	Chloride Penetration Average Depth (mm)		Chloride Ion Diffusion Coefficient (× 10^−12^m^2^/s)	
	28 d	91 d	28 d	91 d
OPC	20.46	10.17	13.3	13.1
ZWR1%	19.46	10.91	13.14	9.51
ZWR3%	16.50	9.82	8.8	6.36
ZWR5%	16.78	9.15	7.39	4.77

**Table 12 materials-13-03288-t012:** Result of accelerated carbonation test.

Name	CO_2_ Penetration Average Depth (mm)				OPC 91D Penetration Ratio (%)
	7d	28d	56d	91d	
OPC	3.62	5.27	5.2	5.8	100
ZWR1%	3.08	4.31	4.88	5.0	86
ZWR3%	2.54	4.12	4.25	4.6	79
ZWR5%	0.78	2.38	2.76	3.0	51

**Table 13 materials-13-03288-t013:** Data of accelerated carbonation test (28 days).

Name	Water Penetration Test (28 d)		
	Weight Difference (g)	Penetration Depth (mm)	Penetration Ratio (%)
OPC	4.58	6.13	100
ZWR1%	2.48	4.93	50
ZWR3%	1.49	3.38	30
ZWR5%	1.1	0.02	20

**Table 14 materials-13-03288-t014:** Data of accelerated carbonation test (91 days).

Name	Water Penetration Test (91 d)		
	Weight Difference (g)	Penetration Depth (mm)	Penetration Ratio (%)
OPC	3.8	5.77	100
ZWR1%	2.9	5.13	80
ZWR3%	1.3	2.68	30
ZWR5%	0.9	0.02	20

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
