# Peer review of "Experimental Study on the Evaluation of Physical Performance and Durability of Cement Mortar Mixed with Water Repellent Impregnated Natural Zeolite"

_materials, 2020, doi:10.3390/ma13153288_

Round 1

Reviewer 1 Report

The article under the title: “Experimental study on for evaluation of physical performance and durability of cement mortar mixed with water repellent impregnated natural zeolite” is in line with the Materials journal. The authors work is the important topic connected with modern  concrete surface treatment technology. The article based on original experimental research. The abstract is sufficiently informative. The organization of the article is appropriate, but, it requires improvements:

- the whole text – references – WRONG NUMERATION OF RFERENCES – please use editor guideline;

- the whole text – references (point after references/brackets);

- the whole text – check grammar, especially capital / small letters application;

- the introduction – “and to reduce CO2 generation . [1,2] he purpose” (please check grammar – „He” insted of ”he”;

- the introduction – wrong format of the references “improving physical properties. [1.9]”;

- the introduction – please explaine NOVEL aspects of your research;

- materials – “washed Korean sand.” – please specify kind of sand not only the origin;

- Figure 2. - 5. and 13.– check grammar – capital letter of the beginning of subtitle should be applied;

- Figure 12. – lack of description for part (b);

- conclusions– please correct the point 6;

- please consider the discussion part, including comparison the received results with the other from literature, especially new one (between 2018 and 2020);

- discussion or conclusions – Please explain - what about prediction for the composites includes more than 5% ZWR?;

- conflicts of Interest – lack of information;

- references – lack of new publication; the newest is from 2017; references MUST be supplemented at the literature between 2018 and 2020, for example: https://www.mdpi.com/1996-1944/13/6/1350/htm

Reviewer 2 Report

I must say that a very interesting topic is described in the manuscript. The durability of cement composites is very important and must be studied. New approaches to protecting cement stone from the harmful effects of the environment are always welcome.
This manuscript is an interesting way to improve the durability of concrete. I have just a few recommendations and questions for authors that need to be discussed.

1) I strongly recommend not to use "unpublished" references.

Choi, H. Y. Study on the construction resources usability of activated hwangtoh, Unpublished doctoral dissertation, Chung Nam National University, Chung Nam, Korea. 2002

2)There are some typographical errors in the text. For example: 28days × 28 days or 40% × 40%.

3) Chapter 2.1: SSA of zeolite 15600cm2/g is written in the text. Which method was used for this measurement - Blaine or BET
I recommend to add the same information in case of OPC.
In opposite density of the zeolite is missing.

4) Chemical compositions in the Tables of 1 and 2 are not complete. The sum is not around 100%. Tab. 1 - 98,29; Tab. 2 - 85,86. I recommend to add column "others" in this cases The table 3 must be in wrong units. Or is the content of Zn aroud 24% correct?

5) The W/B ratio was fixed at 40 % in all cases. But the question is the cement paste consistency. I recommend to add this information, because the zeolite seems to be very fine compare to OPC and except the pozzolanic activity it can works as filler and densify the cement stone naturally. And this should be discussed.

6) I would appreciate to mention the cement paste preparation method briefly. Type of mixer for example. Or just write: cement paste was prepared according to KS (or others) standard.... 

7) Table 8 is in MPa?

8) Fig. 12: The description must be improved. What is b) and what is at figure b)?

9) Fig. 13: The blue line should be better described.

10) I recommend to move the MIP results at the beginning. This method describes microstructure and is crucial for the discussion about durability.

11) The judgement in point 5 of the conclusion about "optimum" is too strong and should be reformulated.

12) I am not native speaker so I don't feel to be fully qualified to judge English language and style. But some sentences were difficult to read for me.

Reviewer 3 Report

Dear authors,

In this paper, the authors evaluate the physical and mechanical performance of cement mortar mixed with water repellent impregnated with natural zeolite. It is an interesting and complete paper with multiple details both in the experimental part and in the subsequent analysis.

The font for Equations must be reduced in accordance with the editing instructions

The figures can be resized in length to better highlight the content

Figure 13 caption in capital letters

Conclusion section

Point 6 please correct

For such a study I recommend for the future the use Design of experiment method..

Round 2

Reviewer 1 Report

The comments are introduced. The text required slight editing corrections.

Reviewer 2 Report

It was my pleasure to review your manuscript. I appreciate you have taken all of my suggestions in to account and accepted most of them.
I have not any other comments.